



# Compound flood impact forecasting: Integrating fluvial and flash flood impact assessments into a unified system

Josias Ritter[1], Marc Berenguer[1], Francesco Dottori[2], Milan Kalas[3], and Daniel Sempere-Torres[1]

[1]Center of Applied Research in Hydrometeorology, Universitat Politècnica de Catalunya, BarcelonaTech, Jordi Girona 1-3 (C4-S1), 08034 Barcelona, Spain
[2]European Commission, Joint Research Centre, Space, Security and Migration Directorate, Via E. Fermi 2749, 21027 Ispra, Italy
[3]Freelance consultant, Sladkovicova 228/8, 01401 Bytca, Slovakia

**Correspondence:** Josias Ritter (ritter@crahi.upc.edu)

**Abstract.** Floods can arise from a variety of physical processes. Although numerous risk assessment approaches stress the importance of taking into account the possible combinations of flood types (i.e. compound floods), this awareness has so far not been reflected in the development of early warning systems: Existing methods for forecasting flood hazards or the corresponding socio-economic impacts are generally designed for only one type of flooding. During compound flood events,

these flood type-specific approaches are unable to identify the overall hazards or impacts. Moreover, from the perspective of the end-users (e.g. civil protection authorities), the monitoring of separate flood forecasts – with potentially contradictory outputs – can be confusing and time-consuming, and ultimately impede an effective emergency response. To enhance the decision support, this paper proposes the integration of different flood type-specific approaches into one compound flood impact forecast. This possibility has been explored by combining the simulations of two impact forecasting methods (representing

fluvial and flash floods) for a recent catastrophic episode of compound flooding: the DANA event of September 2019 in Southeast Spain. The combination of the two methods identified well the overall compound flood extents and impacts reported by various information sources. For instance, the simulated economic losses amounted to about 670 million Euros against 425 million Euros of reported insured losses. Although the compound impact estimates were less accurate at municipal level, they corresponded significantly better to the observed impacts than those generated by the two methods applied separately. This

demonstrates the potential of such integrated approaches for improving the decision support.

## 1  Introduction

Around the globe, floods regularly result in devastating impacts on human society. Between 2008 and 2017, floods claimed on average about 5 000 lives per year (CRED, 2019). With more than one trillion USD over the last four decades, floods accounted for about 40 % of all natural hazard-related economic losses (Munich Re, 2020). Climate change projections suggest that the

frequency and magnitude of floods will increase in many parts of the world over the decades to come (IPCC, 2018). In combination with rising trends in urbanisation and population growth, the impacts of floods on society are expected to increase significantly if no further adaptation measures are adopted (Dottori et al., 2018).



The development of early warning systems (EWSs) is a highly cost-effective way to reduce flood impacts, as they support the coordination of emergency response measures, such as warnings to the population, evacuations, or the installation of temporary

flood barriers (Pappenberger et al., 2015; World Bank, 2010). Flood EWSs generally rely on methods that continuously provide the end-users with forecasts of upcoming flood hazards or impacts (UNISDR, 2006). Typically, these forecasting methods are based on models representing the physical processes that generate floods, which are diverse: The most common flood types include fluvial floods, flash floods, pluvial (urban or surface water) floods, and coastal floods (e.g. European Commission, 2007). Due to the differences in the governing physical processes, forecasting approaches are traditionally designed separately

for the individual flood types (see e.g. Alfieri et al., 2012). Fluvial floods, for instance, develop over days or weeks in large river basins and are most commonly forecasted by coupling weather observations and Numerical Weather Prediction (NWP) with distributed hydrological models. In contrast, flash floods have a more sudden onset (minutes to a few hours) and typically occur in small to medium-sized mountainous catchments. The processes leading to flash floods are usually strongly dominated by extreme rainfall intensities that evolve quickly in time and space, which makes the use of weather radar data attractive for

flash flood monitoring and forecasting (Corral et al., 2019; Georgakakos, 1986; Javelle et al., 2016; Versini et al., 2010).

As of today, the overwhelming majority of flood forecasting approaches focuses on the hazard component of the flood: Methods for fluvial floods (Cloke and Pappenberger, 2009; Jain et al., 2018) or flash floods (Alfieri et al., 2019; Corral et al., 2019; Hapuarachchi et al., 2011) typically forecast peak flows or return periods in the stream network, while pluvial (Henonin et al., 2013; Zanchetta and Coulibaly, 2020) or coastal flood forecasting approaches (Fernández-Montblanc et al., 2019; Kohno

et al., 2018) are mostly designed to predict water levels in the affected areas. For end-users such as civil protection authorities, these flood hazard forecasts form an important part of the decision support in the coordination of emergency response measures. To estimate the expected impacts (e.g. the affected number of people), the end-users combine the hazard forecasts with socio-economic exposure and vulnerability information in the areas at risk. In the current practice, this combination is commonly done based on personal knowledge and experience, or by means of simple GIS-based tools (e.g. Vaz, 2017). However, this

non-automatic procedure of estimating the potential impacts can consume valuable time during approaching events and lead to sub-optimal decisions (Basher, 2006; Merz et al., 2020). For a more effective and faster emergency response, the World Meteorological Organization (WMO, 2015) and the United Nations International Strategy for Disaster Reduction (UNISDR, 2015a) promote the enhancement of the decision support with tools that automatically translate the forecasted hazards into the expected socio-economic impacts.

The general recipe for impact forecasting is similar across flood types (Merz et al., 2020); the forecasted flood hazard is automatically combined with vulnerability and exposure layers, such as population density or land use maps. For fluvial floods, several impact forecasting approaches were developed in recent years (e.g. Bevington et al., 2019; Brown et al., 2016; Cole et al., 2016; Dale et al., 2014; Guimarães Nobre et al., 2020). The Rapid Risk Assessment (RRA; Dottori et al., 2017) predicts with up to 10 days ahead the economic losses, critical infrastructures, and population affected by European rivers with

catchment sizes larger than 500 km$^2$, based on discharge forecasts from the hydrological model LISFLOOD (Roo et al., 2000; Van Der Knijff et al., 2010). As part of the European Flood Awareness System (EFAS), the RRA has been providing for a few years operational decision support to various end-users across the continent. Also with respect to other flood types, progress



in impact forecasting has been made in recent years. For instance regarding flash floods, several approaches are available for predicting impacts in individual catchments or relatively small areas (e.g. Le Bihan et al., 2017; Saint-Martin et al., 2016; Silvestro et al., 2019). The ReAFFIRM method (Ritter et al., 2020a) is the first approach applicable also over larger domains (e.g. at regional or national scale). Based on flash flood hazard nowcasts obtained with the ERICHA system (Corral et al., 2019, 2009), ReAFFIRM estimates in high spatiotemporal resolution (e.g. 25 m and 15 min) the numbers of affected people and critical infrastructures, and the economic losses.

All of the forecasting approaches mentioned above focus on one specific type of flooding. In reality, though, flood events are often the result of a combination of flood types, also referred to as "compound floods" (e.g. Wahl et al., 2015; Zscheischler et al., 2020). The Intergovernmental Panel on Climate Change (IPCC, 2018) defines compound events as "(1) two or more extreme events occurring simultaneously or successively, (2) combinations of extreme events with underlying conditions that amplify the impact of the events, or (3) combinations of events that are not themselves extremes but lead to an extreme event or impact when combined. The contributing events can be of similar (clustered multiple events) or different type(s)". Previous studies on compound floods mostly focused on scenario-based hazard assessments accounting for different combinations of flood types. For instance, Chen et al. (2010) and Apel et al. (2016) applied coupled hydraulic models to simulate combined fluvial and pluvial flooding in urban environments. Similarly, many studies explored the compound hazard from fluvial and coastal flooding, often experienced as a crucial factor during hurricanes (for a review of such approaches, see Santiago-Collazo et al., 2019). To our knowledge, flash floods have so far not been considered in the context of compound flooding.

Although the results of the mentioned hazard assessments stress the importance of taking into account the possible combinations of flood types, this awareness has not yet been addressed by the developers of EWSs. Up to today, forecasting approaches remain flood type-specific. For the end-users of the forecasts, however, a distinction between flood types is secondary. Their main focus is on the provided information on potentially inundated locations and the corresponding impacts, regardless of the underlying flood type. Yet, the end-users' decision-making process is in the current practice usually based on a number of separate flood forecasts (representing the different flood types) that may even show contradictory outputs. This practice is inefficient and might reduce the trust of the end-users in the forecasts. Systems that predict compound events in an integrated way – especially in terms of socio-economic impacts – could significantly improve the decision support (Merz et al., 2020).

This paper proposes the development of a framework that automatically integrates the flood-type specific forecasting approaches into one compound flood impact forecast. A particularly severe episode of compound flooding (the 2019 DANA event in Southeast Spain; Sect. 2) has been taken as an opportunity to explore the possibility of such an integrated system. For this event, we test a simple real-time adapted combination of fluvial flood impact simulations from EFAS RRA (Dottori et al., 2017) with flash flood impact simulations from ReAFFIRM (Ritter et al., 2020a, Sect. 3). The resulting simulated compound impacts for the DANA event are compared to impacts reported by satellite images, flood insurance, civil protection authorities, and the media (Sect. 4). This exploratory study allows for identifying potential opportunities and challenges of combining flood type-specific impact forecasting methods, and future developments required for creating a full compound flood impact forecast encompassing all of the common flood types (Sect. 5).





## 2 The DANA event of September 2019 in Southeast Spain

The south-eastern part of Spain (Fig. 1) is characterised by the hydrometeorological extremes. Almost every year, the region experiences long-lasting droughts as well as torrential rains and floods. To balance the extremes over the course of the year and

compensate for interannual rainfall variabilities, the stream network in the region has been strongly modified through structural interventions. Especially in the basin of the Segura River (19 025 km$^2$ including coastal catchments), the degree of regulation is exceptional: The 33 dams in the basin have an overall capacity of 1 230 Hm$^3$ (CHS, 2020b), which is about 20 % larger than the average yearly rainfall volume in the basin after discounting evapotranspiration (1 027 Hm$^3$; CHS, 2020a). Among various purposes (e.g. public water supply, irrigation, or hydropower generation), the enormous retention capacities in the dams play a

crucial role for flood protection.

From 11 to 15 September 2019, an upper level low-pressure system – in Spain better known as "DANA" or "Gota Fría" (Martín León, 2003) – caused rainfall accumulations of up to 461 mm in 24 h in the region (García et al., 2020). As a result, devastating floods occurred across eight provinces, of which Murcia and Alicante suffered the most severe impacts (for some visual impressions, see the references compiled in CRAHI, 2019). In total, seven people lost their lives and more than 5 000

were evacuated from their homes (Fig. 1). The Spanish Insurance Compensation Consortium (CCS, 2020) recorded private flood insurance claims of more than 450 million Euros, while AON (2019) estimated the overall economic losses from the event to exceed 2.2 billion Euros. The most severe incidents were reported in the floodplain of the lower Segura River (especially in the town of Orihuela), in several coastal towns in the Murcia province, and along some small tributaries of the Jucar River (e.g. the Clariano River; the Jucar River itself did not flood).

One particularly interesting characteristic of this episode is that the most severely affected streams show a high variability in catchment size: While the Segura River has a drainage area of around 15 000 km$^2$ at Orihuela, the catchment of the Clariano River in the Jucar basin is about two orders of magnitude smaller at the most affected town of Ontinyent (160 km$^2$). The large differences in catchment size represent different flood generation mechanisms: on the one hand fluvial flooding, and on the other hand flash flooding. In addition to fluvial and flash floods, the DANA also caused pluvial flooding in several locations,

e.g. in the cities of Alicante, Murcia, Malaga, Madrid, and in Almeria, where one person drowned in a car while crossing a flooded underpass. The combination of fluvial, pluvial, and flash flooding makes this DANA episode a classic example of a compound flood.







**Figure 1.** Summary of the rainfall amounts and impacts of the 2019 DANA event. Adapted from the Emergency Response Coordination Centre of the European Union (ERCC).

## 3 Methods employed for assessing the compound flood impacts

This section describes the two methods that have been employed for simulating the compound impacts of the DANA event.
Fluvial impacts have been estimated using EFAS RRA (Dottori et al., 2017, Sect. 3.1) and flash flood impacts using the ReAFFIRM method (Ritter et al., 2020a, Sect. 3.2). After introducing the two methods separately, the procedure for combining them to a compound flood impact estimation is presented (Sect. 3.3). Table 1 provides an overview of the characteristics and specifications of the employed methods. In this study, both methods have been run based on hydrometeorological observations (rather than forecasts) to minimise external uncertainties and focus on the capabilities and limitations of estimating compound
flood impacts.



**Table 1.** Overview of the characteristics of the two employed methods for assessing the fluvial (EFAS RRA) and flash flood impacts (ReAFFIRM).

| Characteristic | | EFAS RRA | ReAFFIRM |
|---|---|---|---|
| **Scope** | Flood type | Fluvial floods | Flash floods |
| | Stream coverage | Drainage area $\geq$ 500 km$^2$ | 5 km$^2$ $\leq$ Drainage area $\leq$ 2 000 km$^2$ |
| | Domain | Europe | SE Spain<br>(Jucar and Segura basins, 62 000 km$^2$) |
| | Hydrometeorological input<br>(default; this study) | NWP (ECMWF ensemble median);<br>Stream gauge data | Radar rainfall observations and nowcasts;<br>Radar-raingauge-blending |
| | Forecast horizon<br>(default; this study) | up to 10 days;<br>based on observations | up to 6 h;<br>based on observations |
| | Time resolution<br>(default; this study) | 6 h;<br>n/a | 15 min;<br>1 h |
| **Step 1:<br>Hazard<br>estimation** | Base | LISFLOOD (full hydrological model) | ERICHA system (rainfall-based) |
| | Spatial resolution | 5 km | 200 m |
| | Hazard variables | Streamflow, return period | Return period |
| | Return period resolution | [0, 2, 5, 10, 20, 50, 100, 200, 500, 1000] years | [0, 2, 5, 10, 25, 50, 100, 200, 500] years |
| **Step 2:<br>Flood depth<br>estimation** | Flood maps used | EFAS flood maps (Dottori et al., 2017) | National official flood maps<br>(in Spain: IGN, 2020a) |
| | Spatial resolution | 100 m | 1 m, upscaled to 25 m |
| | Return period resolution | [10, 20, 50, 100, 200, 500] years | [10, 50, 100, 500] years |
| **Step 3:<br>Impact<br>estimation** | Categories | Population in the flooded areas,<br>economic losses [€],<br>affected critical infrastructures (CIs) | Population in the flooded areas,<br>economic losses [€],<br>affected critical infrastructures (CIs) |
| | Spatial resolution | 100 m | 25 m |
| | Spatial aggregation<br>(default; this study) | NUTS 2 or NUTS 3 regions;<br>Municipalities | Municipalities;<br>Municipalities |


### 3.1 Fluvial flood impacts: The EFAS Rapid Risk Assessment (RRA)

This section briefly describes the EFAS RRA (for full detail, see Dottori et al., 2017), which has been used to estimate the fluvial component of the flood impacts. The method consists of three steps (see also Table 1):

1. Hazard estimation: Real-time discharge observations and NWP forecasts are used as input to the LISFLOOD hydrological model. Every 6 h, the model simulates the streamflow over the European drainage network in 5 km resolution.

2. Flood depth estimation: The streamflow simulated by LISFLOOD is transformed into flood extents and depths in 100 m resolution. This is done based on a pre-calculated inventory of flood maps of several discharge return periods (Table 1), covering rivers with catchments larger than 500 km$^2$. It is important to note that the flood maps were generated at pan-European scale and therefore have certain limitations in resolution and accuracy (for an evaluation in Spain, see Dottori et al., 2021).

3. Impact estimation: The simulated flood extents and depths are combined with exposure and vulnerability datasets to assess the socio-economic impacts. Three impact categories are included: Affected population (using the population density map of Freire et al., 2016), critical infrastructures (using the infrastructure database of Giovando et al., 2020), and direct economic losses (using CORINE land cover and the depth-damage functions of Huizinga et al., 2017). The impacts are automatically aggregated for the administrative regions (NUTS 2 or NUTS 3) to provide a concise summary of the outputs.

After this general description of EFAS RRA, we focus now on the particularities of the application of the method in this study. We have decided to substitute the LISFLOOD discharge simulations (step 1 of the method) with the discharges measured by the stream gauges of the Hydrographic Confederation of the Segura (CHS, 2021). The reasoning behind this decision is described in the following: The recent long-term validation of LISFLOOD shows for the Segura one of the lowest performance scores of all the European catchments (Mazzetti and Harrigan, 2020), which has been attributed to two main sources: Firstly, due to the high degree of flow regulation by dams and other hydraulic structures in the Segura basin (see Sect. 2). The rules on which dam operators base their release decisions are typically unknown (e.g. Nazemi and Wheater, 2015; Ritter et al., 2020b), usually hindering an adequate representation of the effects of dams in hydrological models. Secondly, large parts of Southeast Spain, and in particular the Segura basin, are situated on a highly karstic topography (Goldscheider et al., 2020), in which hydrological simulations generally show high uncertainties (Hartmann et al., 2014). To substitute the LISFLOOD simulations for the DANA event, we have included discharge data from eight stream gauges along the Segura River (Fig. 2). The discharge observations have been connected to step 2 of the method as follows: In each river reach, the flood map that corresponded most closely to the measured peak flow has been selected from the set of EFAS flood maps (Fig. 2). The resulting mosaic of flood maps represents the maximum of simulated flood extents and depths over the full event duration (11–14 September 2019). To simulate the corresponding flood impacts, the maximum flood depths have been combined with exposure and vulnerability layers. In this last step, the default configuration of EFAS RRA has been applied (see step 3 of the method); however, the impact aggregation has been done at the level of municipalities to enable a more detailed analysis (Table 1).



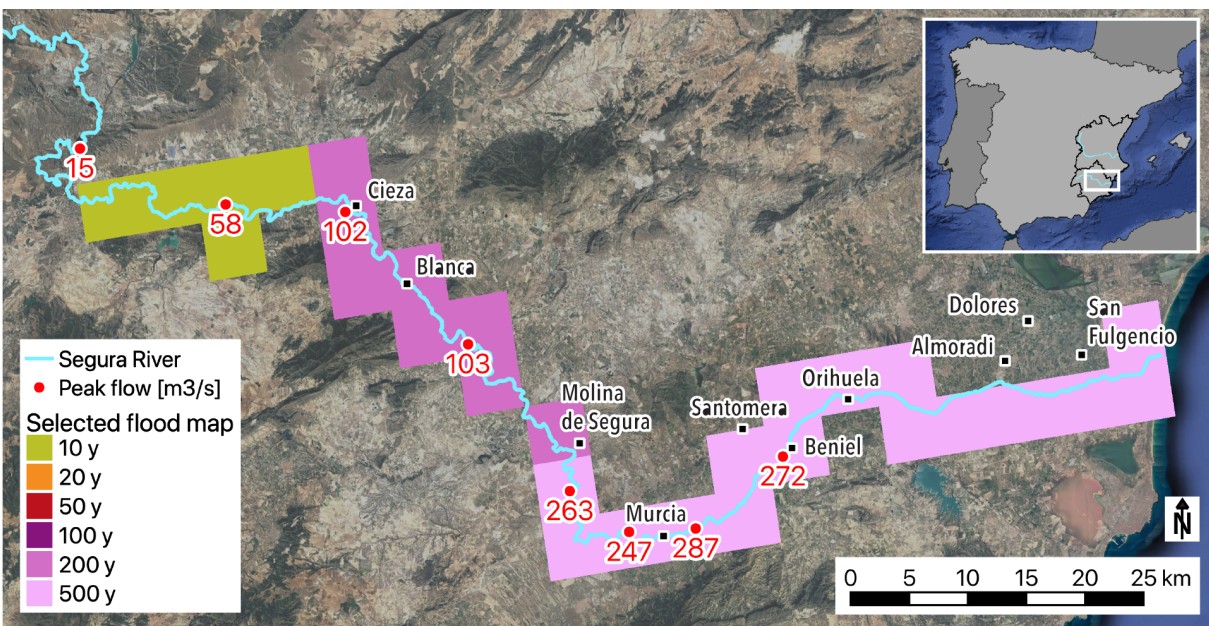

**Figure 2.** Measured peak flows [m³/s] in the Segura River during the DANA event (CHS, 2021) and the corresponding selection of EFAS flood maps along the LISFLOOD drainage network (represented by the 5 km-grid cells). Map data ©Google Earth 2015.

## 3.2 Flash flood impacts: The ReAFFIRM method

The ReAFFIRM method (for full detail, see  Ritter et al., 2020a) has been used in this study to estimate the flash flood-induced impacts of the DANA event. The method assesses impacts originating from streams with catchment areas between 5 and 2 000 km². Similar to EFAS RRA, also ReAFFIRM consists of three main steps (Table 1):

1. First of all, a flash flood hazard module (the ERICHA system; Corral et al., 2009, 2019) uses weather radar observations to estimate the hazard return periods over a gridded drainage network.

2. Then, a flood map module translates the estimated hazard return periods into high-resolution flood extents and depths, based on the official flood maps created in the framework of the EU Floods Directive (European Commission, 2007).

3. Finally, an impact assessment module employs several layers of socio-economic exposure and vulnerability in the flooded areas to estimate the flash flood impacts in three categories: population in the flooded areas, economic losses, and affected critical infrastructures (CIs).

Initially applied and tested in Catalonia (Northeast Spain; Ritter et al., 2020a), the ReAFFIRM method has now been applied in the hydrographic demarcations (hereafter referred to as basins) of the Jucar and Segura rivers, covering an overall area of almost 62 000 km² (Fig. 3). More than 92 % of the economic losses from the DANA event occurred within this domain (CCS, 2020). The configuration of ReAFFIRM and the datasets used in this region are described in the following (see also Table 1:





i. Rainfall inputs: As rainfall inputs for step 1 of ReAFFIRM, we have used the radar composites from OPERA (Operational Program for the Exchange of weather RAdar information; (www.eumetnet.eu/opera), which are produced in real time and resolutions of 2 km and 15 min. Although improved by a chain of real-time adjustment algorithms (Saltikoff et al., 2019; see also Park et al., 2019), the OPERA rainfall products significantly underestimated the observed rainfall during the DANA event (Fig. 3a–b). To reduce the bias in the rainfall inputs for the analysis of the event, we have applied the radar-raingauge-blending technique proposed by Velasco-Forero et al. (2009; see also Cassiraga et al., 2020), using the raingauge measurements of the Spanish State Meteorological Agency (AEMET) with an hourly time step (as also done by (Ritter et al., under review). The resulting improved rainfall estimates are shown in Fig. 3c–d. It can be seen that the largest rainfall amounts were observed near the severely affected towns of Orihuela, Los Alcazares, and Ontinyent.

ii. Flash flood hazard module (step 1): The ERICHA flash flood hazard system has been set up on the base of a topography grid in 200 m resolution (IGN, 2020a). The exceeded return period in each cell of the gridded drainage network is computed by comparing the observed basin-aggregated rainfall to thresholds derived from the historical raingauge analysis of Ministerio de Fomento (1999). To estimate the critical rainfall duration for the upstream drainage area of each cell, the Kirpich (1940) time of concentration formula has been used.

iii. Flood map module (step 2): The official flood maps in the domain are freely provided by the Spanish National Geographic Institute (IGN, 2020a). Flood extent maps are available for all areas in which "potential significant flood risks exist or might be considered likely to occur" (European Commission, 2007). For around 74 % of the area covered by the flood extent maps, also flood depth data is available in 1 m resolution. The flood depths have been upscaled to 25 m (the resolution used by ReAFFIRM). For the flood extents in which flood depth data was unavailable, a uniform flood depth of 0.5 m has been assumed (following Ritter et al., 2020a).

iv. Impact assessment module (step 3): For estimating the population in the flooded areas, the population density map of Freire et al. (2016) has been interpolated to 25 m resolution, and the Spanish national land use dataset SIOSE (reference year 2014; IGN, 2020b) has been applied as a filter to keep population density values only in the populated land uses (residential, industrial, and commercial). The SIOSE land use dataset has also been used for estimating the economic losses, combined with the depth-damage functions from Huizinga et al. (2017) adjusted for Spain. Locations of CIs (education facilities, health facilities, and mass-gathering sites) were extracted from OpenStreetMaps in the framework of the project "Global Exposure Data for Risk Assessment" (Giovando et al., 2020).

The default configuration of ReAFFIRM simulates lower and upper bounds of flood extents and impacts (Ritter et al., 2020a). Throughout this paper, the illustrations of the simulated flood extents in the maps refer to the upper bound of flood extents. The impact estimates listed in the result tables represent the mean values of the lower and upper bounds.



**Figure 3.** Total rainfall accumulations (11–14 September 2019) for the real-time adjusted OPERA radar (a and b) and for the result of the radar-raingauge-blending used throughout this paper (c). d) Performance of the radar-raingauge-blending evaluated by means of ("leave-one-out") cross validation. In panels a and c, the raingauge accumulations and their locations are represented by the circles, and the black lines are the limits of the catchment administrations of the Jucar (north) and the Segura (south).



### 3.3 Compound flood impact estimation

The proposed approach to generate the compound flood impact estimates automatically combines the impacts of fluvial floods with those of flash floods, estimated by EFAS RRA and ReAFFIRM, respectively. Fluvial floods and flash floods mostly occur in different parts of the stream network (fluvial floods in large rivers and flash floods in smaller streams). Hence, for this particular combination of flood types, the compound flood impacts have been approximated as the sum of impacts of the individual flood types. However, to avoid biases, the following consideration has been made:

EFAS RRA estimates the impacts in the rivers with catchment areas larger than $500 \, km^2$, while ReAFFIRM focuses on the smaller catchments with $5$–$2\,000 \, km^2$ size (Table 1). This means that in the streams with catchment areas of $500$–$2\,000 \, km^2$, both fluvial and flash flood impacts can be detected at the same time. Moreover, also at confluences of large rivers and small tributaries, the simulated impacts of the two methods can overlap. To avoid a systematic overestimation of impacts, we have decided to select in such situations the results from EFAS RRA, since past studies have shown that the impact estimates of
ReAFFIRM are subject to increased uncertainties near large rivers (Ritter et al., 2020a, 2021).

This decision has enabled a straightforward combination of the two impact assessments: Wherever EFAS RRA detects (fluvial) flood extents, the (flash) flood extents and impacts simulated by ReAFFIRM are automatically removed. Then, the (unchanged) fluvial and the (cropped) flash flood extents and impact estimates are instantly merged, resulting in a continuous coverage for the catchments larger than $5 \, km^2$.

Finally – as also done in the two individual methods (Table 1) – the resulting compound flood extents and impacts are aggregated at the level of municipalities.

### 4 Results

This section presents the impacts of the DANA event simulated separately by the two methods (Sect. 4.1 and 4.2) and by the compound flood impact estimation (Sect. 4.3). Since the results of EFAS RRA have been generated based on measured peak
flows (instead of using the default 6-hourly discharge simulations), they represent the simulated maximum impacts over the full event duration (i.e. one set of outputs for the entire event; Sect. 3.1). Although the results from ReAFFIRM in this study have been generated at hourly resolution (Sect. 3.2) we present in this section also for ReAFFIRM the aggregated impacts over the full event duration to enable a comparison with EFAS RRA's results and post-event impact observations.

The simulation results are compared with impacts reported by the media (compiled in CRAHI, 2019) and the Spanish
Directorate-General for Civil Protection and Emergencies (DGPCE, 2019). Furthermore, we compare the simulated economic losses to a database of flood insurance claims provided by the Spanish Insurance Compensation Consortium (CCS, 2020). This database reports for each municipality the claimed losses and therefore provides valuable information on the spatial distribution of losses over the domain. However, it contains only the insured and claimed losses in the private, industrial, and commercial sectors – agriculture and public infrastructures are not included. For instance, during the DANA event claimed losses of 206.5
million Euros were recorded in the insurance database in the 45 municipalities of the Murcia province (CCS, 2020), whereas the overall economic losses in the province (including all sectors) amounted to about 590 million Euros (Arbáizar-Barrios,





2019). Based on these numbers, a rough factor of 2.5–3 can be assumed to convert the values in the insurance claim database into overall economic losses. This rough factor helps to put the insured losses into perspective when they are confronted with the values of overall economic losses estimated by EFAS RRA and ReAFFIRM.

### 240  4.1  Fluvial flood impacts estimated by EFAS RRA

This section presents the simulated fluvial flood impacts of the DANA event. Using the stream gauge data along the Segura River as input (Sect. 3.1), EFAS RRA estimated the fluvial flood extents shown in blue and purple in Fig. 4: In the upstream part of the Segura, several minor inundations near the river were identified. In contrast, in the lowlands downstream of the city of Murcia, the simulated fluvial flood extents cover vast areas, reaching up to about 8 km from the river channel (Fig. 4). This

general image corresponds well to the situation described by the authorities after the DANA event: the Segura overflowed the flood protection structures in several locations downstream of Murcia and widely inundated the flat terrain (DGPCE, 2019).

Based on the simulated fluvial flood extents and depths (Fig. 4), EFAS RRA identified 28 381 people and 17 CIs in flooded areas, and economic losses of 422.8 million Euros (Table 2). These simulated fluvial flood impacts correspond relatively well to the reported overall impacts (although the reported numbers include also impacts induced by other flood types that EFAS

RRA is not designed to detect). The simulated impacts are distributed over 36 municipalities along the Segura River, of which 10 are listed in Table 2.

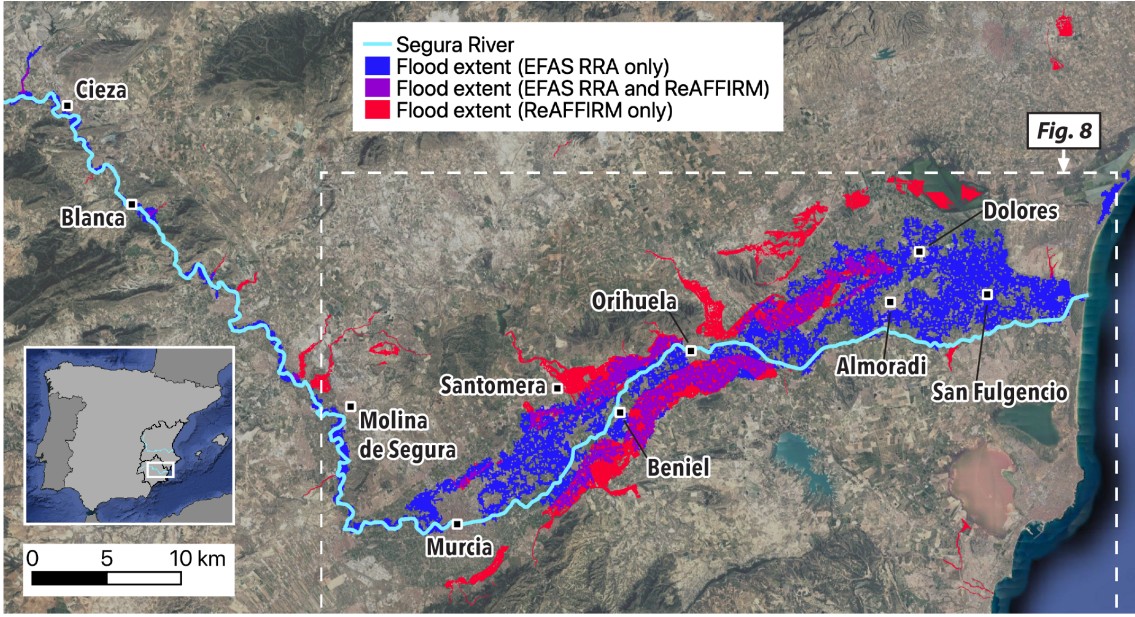

**Figure 4.** Fluvial flood extents simulated by EFAS RRA (blue) and flash flood extents simulated by ReAFFIRM (red). In the locations where the flood extents simulated by the two methods overlap (purple), the flash flood extents and impacts simulated by ReAFFIRM are automatically removed to avoid a systematic overestimation of the compound impacts (see Sect. 3.3). The dashed rectangle indicates the area displayed in Fig. 8. Map data ©Google Earth 2015.





**Table 2.** Summary of simulated fluvial flood impacts (EFAS RRA) and reported flood impacts (CRAHI, 2019; CCS, 2020; DGPCE, 2019) in selected municipalities. Critical infrastructures (CIs) are abbreviated as health facilities (HF), education facilities (EF), and mass-gathering sites (MG).

| MUNICIPALITY | SIMULATED IMPACTS | | | | REPORTED IMPACTS | |
| --- | --- | --- | --- | --- | --- | --- |
| | Flooded area [ha] | Population in flooded areas | Losses [M€] | CIs | Insured losses [M€] | Other |
| SEGURA & JUCAR BASINS | 16 011 | 28 381 | 422.8 | 4 EF, 3 HF, 10 MG | 425.2 | 5 fatalities; min. 6 260 evacuated |
| Murcia | 3 675 | 20 191 | 248.8 | 4 EF, 5 MG | 35.2 | evacuations |
| Orihuela | 3 809 | 2 417 | 39.0 | 3 HF, 3 MG | 105.4 | 2 fatalities; 150 rescued; 70 evacuated |
| Blanca | 194 | 1 402 | 15.8 | 2 MG | 0.4 | 80 evacuated |
| Almoradi | 983 | 986 | 17.7 | | 15.1 | evacuations |
| Beniel | 285 | 554 | 14.3 | | 4.0 | 14 evacuated |
| Dolores | 991 | 348 | 14.1 | | 15.0 | 1 fatality; evacuations |
| San Fulgencio | 989 | 211 | 25.7 | | 2.7 | evacuations (about 10 families) |
| Santomera | 325 | 55 | 9.3 | | 2.1 | min. 2 200 evacuated (dam emergency) |
| Cieza | 341 | 49 | 18.6 | | 3.4 | 56 evacuated |
| Molina de S. | 285 | 2 | 0.2 | | 10.7 | 40 evacuated |

In many of the municipalities flooded by the Segura downstream of Murcia, the quantitative impact estimates are approximately in line with the reported impacts (e.g. in Almoradi, Dolores, Beniel, and Santomera; Table 2). However, the impacts were clearly underestimated in the most severely affected municipality of Orihuela, since a significant part of the impacts in
this location were caused by flash floods in small tributaries of the Segura (e.g. the two fatalities listed in Table 2). Similarly as in Orihuela, flash floods in small tributaries were also responsible for a large share of the impacts in Molina de Segura, explaining the impact underestimation by EFAS RRA also in this municipality (Table 2). In contrast, the impacts in the municipality of Murcia have been significantly overestimated (Table 2): Fluvial flooding was reported in the rural areas upstream and downstream of the city of Murcia, but not in the city centre, where the high local flood protection levels prevented the
Segura from flooding the urban areas (CRAHI, 2019). Since the pan-European flood maps used by EFAS RRA do not account for such local defence structures (Dottori et al., 2021), the flood extents in the city of Murcia and the corresponding impacts were significantly overestimated (Fig. 4 and Table 2). Similar effects leading to overestimated impacts were observed in Cieza and Blanca in the upstream part of the Segura, and in San Fulgencio close to the river mouth (Fig. 4 and Table 2).





## 4.2 Flash flood impacts estimated by ReAFFIRM

This section presents the flash flood impacts simulated by ReAFFIRM for the DANA event. Overall, ReAFFIRM estimated in the Segura and Jucar basins 43 091 people in flooded areas, 290.2 million Euros in economic losses, and 16 affected CIs (Table 3). The impacts are spread over a total of 100 municipalities, indicated in Fig. 5a in the colours red (flood affecting population; 41 municipalities), orange (flood causing economic losses but not affecting population; 38 municipalities), and yellow (flood not affecting population or assets; 31 municipalities). A first visual inspection reveals that the locations of the

simulated impacts (Fig. 5a) correspond very well to those of the reported impacts (Fig. 5b): Simulated impacts appear in most of the municipalities where people were rescued or evacuated. Furthermore, ReAFFIRM identified impacts in the two municipalities with flash flood-related fatalities (Table 3), although the signal is small in Caudete, where two persons died in their vehicle on a flooded country road (CRAHI, 2019).

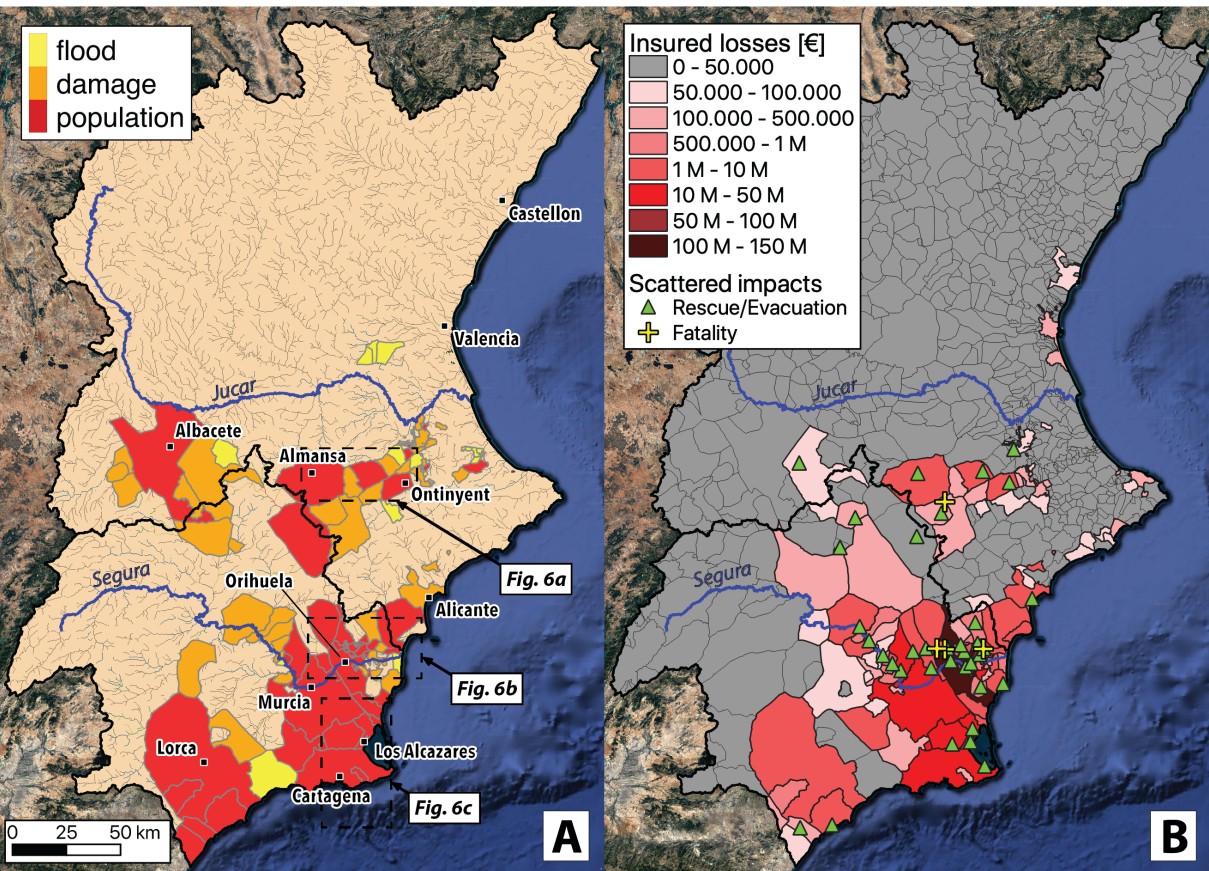

**Figure 5.** a) Flash flood impacts simulated by ReAFFIRM (11–14 September 2019). b) Reported flood impacts: Insured economic loss by municipality (CCS, 2020), and locations of fatalities, rescues, and evacuations gathered by the media and the civil protection authorities (CRAHI, 2019; DGPCE, 2019). Map data ©Google Earth 2015.





**Table 3.** Summary of simulated flash flood impacts (ReAFFIRM) and reported flood impacts (CRAHI, 2019; CCS, 2020; DGPCE, 2019) in selected municipalities (11–14 September 2019; corresponding to Fig. 5). The critical infrastructures (CIs) are abbreviated as education facilities (EF), health facilities (HF), and mass-gathering sites (MG).

| MUNICIPALITY | SIM. HAZARD | | SIMULATED IMPACTS | | | REPORTED IMPACTS | |
| --- | --- | --- | --- | --- | --- | --- | --- |
| | Max. T [years] | Flooded area [ha] | Population in flooded areas | Losses [M€] | CIs | Insured losses [M€] | Other |
| SEGURA & JUCAR BASINS | 500 | 26 161 | 43 091 | 290.2 | 2 EF, 4 HF, 10 MG | 425.2 | 5 fatalities; min. 6 260 evacuated |
| Cartagena | 500 | 5 920 | 8 740 | 67.0 | 2 EF, 1 HF, 1 MG | 17.9 | min. 95 evacuated |
| Los Alcazares | 200 | 835 | 6 951 | 19.2 | 2 MG | 60.4 | evacuations |
| San Javier | 200 | 940 | 4 934 | 21.1 | 1 HF, 1 MG | 26.0 | evacuations |
| Torre-Pacheco | 100 | 3 094 | 4 248 | 24.1 | 2 HF | 21.0 | evacuations |
| Orihuela | 500 | 3 549 | 2 236 | 71.5 | 4 MG | 105.4 | 2 fatalities; 130 rescued; 70 evacuated |
| Almansa | 100 | 571 | 2 166 | 11.5 | | 1.5 | evacuations |
| Santomera | 200 | 643 | 950 | 9.3 | | 2.1 | min. 2 200 evacuated (dam emergency) |
| Mogente | 100 | 40 | 77 | 1.1 | | 1.4 | evacuations |
| Ontinyent | 10 | 74 | 76 | 3.6 | | 6.4 | 40 rescued; 150 evacuated |
| Caudete | 10 | 8 | 0 | 0.1 | | 0.5 | 2 fatalities |

ReAFFIRM detected the most significant flash flood impacts in the three parts of the domain indicated by the dashed boxes
in Fig. 5a and shown more closely in Fig. 6:

In the Jucar basin, ReAFFIRM detected significant impacts along the rivers Cañoles and Clariano (Fig. 6a). For the Cañoles River in Almansa (365 km²), the ERICHA system estimated a return period of T = 100 years, resulting in somewhat overestimated impacts in this municipality (Table 3). The real flood peak in Almansa was probably lowered by an upstream dam not taken into account by the ERICHA system (see the dam's location in Fig. 6a). Further downstream in Mogente (862 km²),
the return period of T = 100 years seems to be in line with the observed flood magnitude (CRAHI, 2019), and the relatively low simulated economic losses of 1.1 million Euros in this rural municipality correspond well to the insured losses (Table 3). Although the town of Ontinyent (Fig. 6a) experienced unprecedented flooding from the Clariano River (CRAHI, 2019), the estimated return period in this location is only 5–10 years. This low hazard estimate stems from a rainfall underestimation in the small catchment (160 km²): A few kilometres upstream of the town, three local raingauges recorded 333–344 mm on the
day of the flood (12 September 2019; AVAMET, 2019), whereas the radar (blended with the national raingauges) estimated for the same day only 233–250 mm in the raingauge locations. This rainfall underestimation propagated down to the impact estimates in Ontinyent (Table 3).





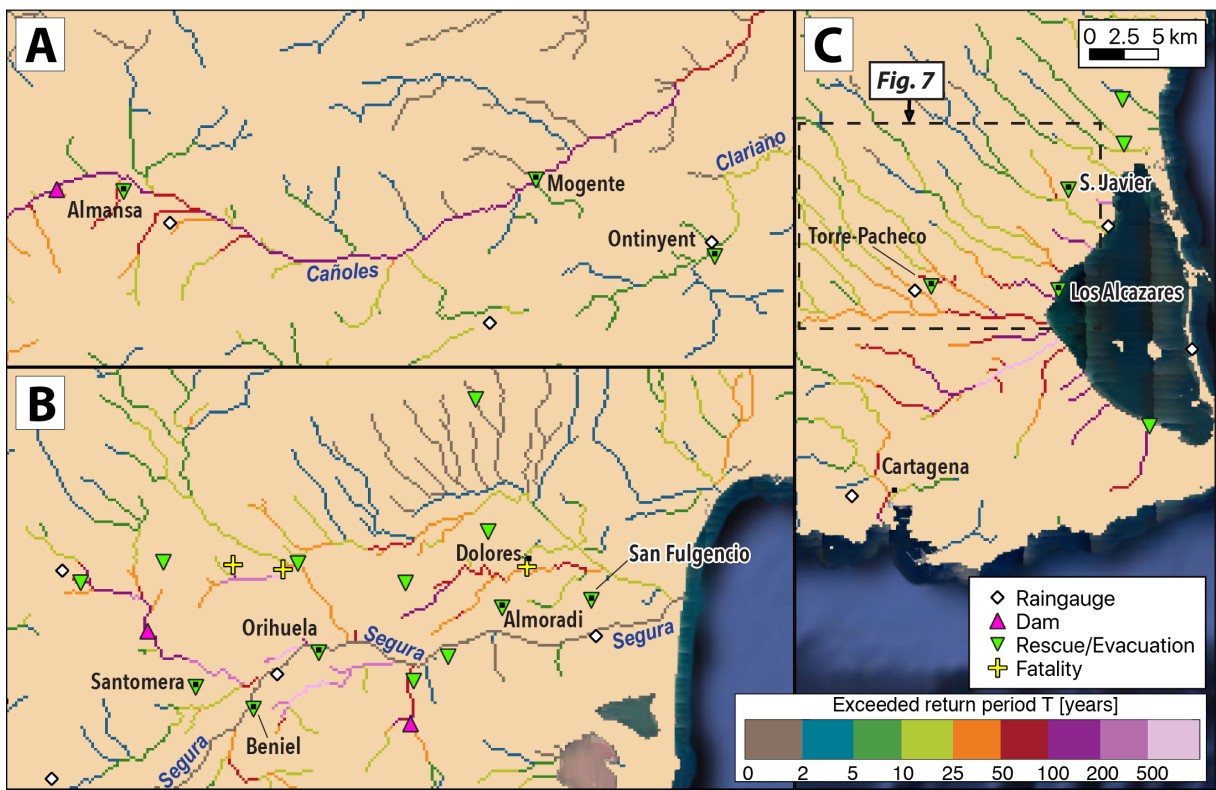

**Figure 6.** Maximum flash flood hazard level (11–14 September 2019) simulated by the ERICHA system in the most severely affected parts of the domain. The locations of panels a–c are indicated in Fig. 5a.

Around the town of Orihuela (Fig. 6b), the ERICHA system estimated return periods of up to $T = 500$ years in the small tributaries of the Segura River, resulting in significant flood extents simulated by ReAFFIRM (Fig. 4). These results seem to be in line with the reported fatalities and evacuations along these tributaries (Fig. 6b). The overall impacts, however, seem to be underestimated in Orihuela (Table 3). This is due to the fact that – in addition to the flash floods – also exceptional fluvial flooding from the Segura River affected the municipality (Sect. 4.1; DGPCE, 2019). For the Segura River itself, no flash flood hazard has been estimated, since the catchment area of around $15\,000\,\text{km}^2$ at Orihuela is far above the limit of the ERICHA system ($2\,000\,\text{km}^2$, see Sect. 3.2).

Severe flash floods also affected the south-eastern part of the domain (Fig. 6c). The ERICHA system identified return periods of $T = 100$–$200$ years in the municipalities of Torre-Pacheco, Los Alcazares, and S. Javier, in ephemeral streams with flat catchments areas in the order of $10$–$100\,\text{km}^2$ (Fig. 7a). These high return periods correspond well to the exceptional reported impacts in these three municipalities (Table 3). In Los Alcazares and Torre-Pacheco (Fig. 6c), the flood extents during the event were recorded by the satellite-based Copernicus Rapid Mapping service (ERCC, 2019, Fig. 7b). The satellite image in Torre-Pacheco dates from the morning of 13 September 2019 (i.e. only a few hours after the flood peak). It can be seen that the simulated flood extent in the south-eastern part of the municipality corresponds reasonably well to the recorded flood extent





(Fig. 7b). In the rural lands west of the town, ReAFFIRM underestimated the flood extents since the employed flood maps did not include the small streams in this area (Fig. 7a). Overall, however, the simulated impacts in Torre-Pacheco correspond well to the reported (Table 3). Also in Los Alcazares, the simulated and observed flood extents are similar (Fig. 7b). However, the flood extent observed in this location was obtained from satellite observations taken five days after the peak of the event, suggesting that the real flood extent in Los Alcazares was significantly larger than recorded. Furthermore, the Civil Protection reported that in reality the municipality of Los Alcazares was inundated in its entirety (DGPCE, 2019). This means that ReAFFIRM underestimated the flood extent in Los Alcazares (and thus the impacts; Table 3). One reason for this underestimation is that – similarly as in Torre-Pacheco – the employed flood maps cover only part of the municipality (Fig. 7a). In flat areas such as this part of the domain, the flood maps are subject to high uncertainties due to the increased complexity of the underlying hydraulic simulations. Similar uncertainties in flat terrain also appeared further south in the municipality of Cartagena (Fig. 6c), where the flood maps of $T = 50$ years show widespread flooding along a few streams. This resulted in overestimated flood extents and impacts in the city centre of Cartagena and a few smaller towns upstream that suffered lower impacts in reality (Table 3).





**Figure 7.** a) Maximum ERICHA flash flood hazard (11–14 September 2019) and official flood maps used for the simulation of the flood extents in Torre-Pacheco and Los Alcazares (for the location of the shown area, see Fig. 6c). b) Comparison of the simulated flash flood extents with satellite observations in Torre-Pacheco (13 September 2019 10:50 UTC) and in Los Alcazares (18 September 2019 10:51 UTC). Map data ©Google Earth 2015.



### 4.3 Estimated compound flood impacts

To estimate the compound flood extents and impacts, the simulation results of EFAS RRA (Sect. 4.1) and ReAFFIRM (Sect. 4.2) have been combined by following the simple procedure described in Sect. 3.3. The resulting compound flood extents in the areas along the Segura River are illustrated in Fig. 8 in red and blue. Also in this part of the domain, we have compared the simulated flood extents to satellite observations from the Copernicus Rapid Mapping Service (ERCC, 2019, Fig. 8). The satellite image in this location dates from 14 September 2019 17:52 UTC, i.e. about 30 h after the measured discharge peak in the Segura passed the most severely affected town of Orihuela (CHS, 2021). Even though the flood had already mostly receded at that time, many areas located several kilometres far from the Segura still appear inundated in the satellite observations. The locations of these flooded patches indicate how far the water from the Segura must have reached during the peak of the event. From around Beniel to the river mouth, the flooded patches line up relatively well with the outlines of the simulated flood extents (Fig. 8), which indicates a good general correspondence between the simulated and the real flood extents in the downstream part of the Segura (although the flood extents were somewhat underestimated north-east of Dolores). The inundated areas north-west of Dolores (Fig. 8) originated not from the Segura but from a small tributary catchment and were correctly identified by ReAFFIRM (see the estimated return periods of T = 10–50 years in this location in Fig. 6b). Along the other tributaries of the Segura affected by flash floods, the inundations had already fully receded at the time of the satellite image acquisition. For instance, the flood peak in the stream north of Orihuela, where the two fatalities occurred (Fig. 6b), was observed on 13 September 2019 at 08:15 UTC (CRAHI, 2019), about 34 h before the satellite image was recorded (Fig. 8). At 08:00 UTC, ERICHA detected a return period of T = 500 years in this stream (Fig. 6b), indicating a good timing of the hazard signal in this location.

Over the two analysed river basins, the combination of EFAS RRA and ReAFFIRM identified 70 278 people and 31 CIs in flooded areas, and 668.9 million Euros of economic losses (Table 4). These numbers correspond relatively well to the reported impacts over the domain (Table 4). When analysing the results at the municipality level, the uncertainties affecting the compound impact estimates are more apparent (see e.g. in Table 4 the large differences between simulated and insured economic losses in Los Alcazares or Murcia).



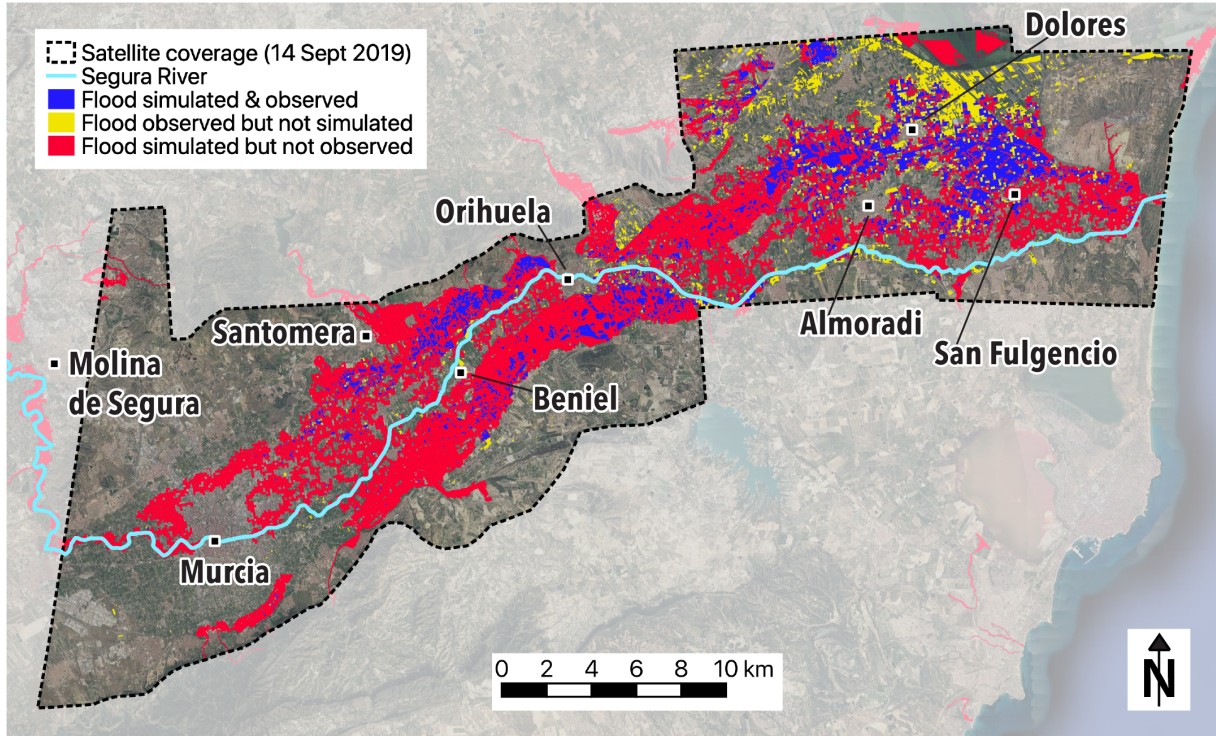

**Figure 8.** Comparison of the simulated compound flood extents with the satellite observation on 14 September 2019 17:52 UTC. The location of the shown area is indicated by the dashed box in Fig. 4. Map data ©Google Earth 2015.

To clearly illustrate the complementarity of the two methods, we have analysed in detail the 15 municipalities with more than 10 million Euros in either simulated or insured losses (Fig. 9). As expected, EFAS RRA detected the losses induced by

fluvial flooding along the Segura River (e.g. in Almoradi, Dolores, Beniel, or Cieza), whereas ReAFFIRM identified the losses in the municipalities that experienced flash floods (e.g. Los Alcazares, San Javier, Torre-Pacheco, Cartagena, or Almansa; Fig. 9). While the individual impact assessments of the two methods detected only the impacts induced by the specific flood type they are designed for, the compound impact estimation identified significant losses in all of the severely affected municipalities in the domain (Fig. 9). In other words, the false negatives (misses) in these 15 municipalities have been reduced to zero through

the combination of the two methods. However, the false positives (false alarms) caused by the uncertainties in the individual methods cascaded down to the compound impact estimates. For instance, the significant overestimations of losses caused by EFAS RRA in Blanca (see Sect. 4.1) or by ReAFFIRM in Almansa (see Sect. 4.2) also appear in the simulated compound losses (Fig. 9).

In only 3 of the 15 analysed municipalities, significant losses were simulated by both EFAS RRA and ReAFFIRM (Orihuela,

Murcia, and Santomera; Fig. 9). These three municipalities were also in reality affected by combinations of fluvial and flash floods. In Orihuela, the estimated compound losses are lower than the real losses, since the combination of the methods identified only inundations of the agricultural lands and settlements surrounding the town but not in the severely affected town





**Table 4.** Summary of simulated and reported compound flood impacts in the 15 municipalities with more than 10 million Euros in simulated or insured losses (corresponding to Fig. 9). The critical infrastructures (CIs) are abbreviated as education facilities (EF), health facilities (HF), and mass-gathering sites (MG).

| MUNICIPALITY | SIMULATED COMPOUND IMPACTS | | | | REPORTED IMPACTS | |
| --- | --- | --- | --- | --- | --- | --- |
| | Flooded area [ha] | Population in flooded areas | Losses [M€] | CIs | Insured losses [M€] | Other |
| SEGURA & JUCAR BASINS | 38 985 | 70 278 | 668.9 | 6 EF, 7 HF, 18 MG | 425.2 | 5 fatalities; min. 6 260 evacuated |
| Orihuela | 5 380 | 4 043 | 74.7 | 3 HF, 5 MG | 105.4 | 2 fatalities; 130 rescued; 70 evacuated |
| Los Alcazares | 835 | 6 951 | 19.2 | 2 MG | 60.4 | evacuations |
| Murcia | 4 584 | 21 038 | 253.4 | 4 EF, 5 MG | 35.2 | evacuations |
| San Javier | 940 | 4 934 | 21.1 | 1 HF, 1 MG | 26.0 | evacuations |
| Torre-Pacheco | 3 094 | 4 248 | 24.1 | 2 HF | 21.0 | evacuations |
| Cartagena | 5 920 | 8 740 | 67.0 | 2 EF, 1 HF, 1 MG | 17.9 | min. 95 evacuated |
| Almoradi | 1 002 | 986 | 17.7 | | 15.1 | evacuations |
| Dolores | 993 | 348 | 14.2 | | 15.0 | 1 fatality; evacuations |
| Molina de S. | 523 | 106 | 7.1 | 1 MG | 10.7 | 40 evacuated |
| Beniel | 440 | 566 | 14.7 | | 4.0 | 14 evacuated |
| Cieza | 344 | 49 | 18.6 | | 3.4 | 56 evacuated |
| San Fulgencio | 1 013 | 213 | 25.7 | | 2.7 | evacuations (about 10 families) |
| Santomera | 830 | 997 | 18.2 | | 2.1 | min. 2 200 evacuated (dam emergency) |
| Almansa | 571 | 2 166 | 11.5 | | 1.5 | evacuations |
| Blanca | 199 | 1 402 | 15.9 | 2 MG | 0.4 | 80 evacuated |

centre (CRAHI, 2019, Fig. 8). One reason for the flood extent underestimation in this location might by the high uncertainty of the EFAS flood maps in urban areas due to limitations of the underlying elevation data (see Dottori et al., 2021). In the
municipality of Murcia, EFAS RRA significantly overestimated the fluvial flood impacts (see Sect. 4.1) and this overestimation propagated down to the compound impact estimates (Fig. 9). Also ReAFFIRM identified impacts in Murcia (Fig. 9) and it could be verified that flash floods occurred in some of the estimated impact locations (CRAHI, 2019). The impacts in Santomera were mostly induced by fluvial flooding from the Segura River, as correctly identified by EFAS RRA (Fig. 9). The flash flood impacts in this municipality were overestimated, since the real discharge peak in the affected tributary was significantly lowered by
a dam not taken into account by the ERICHA system (see the location of the dam in Fig. 6b). The dam's buffer capacity prevented a catastrophic flash flood in this tributary (Arbáizar-Barrios, 2019), but the critically high storage level required the evacuation of more than 2 200 people in the town of Santomera, situated between the dam and the confluence with the Segura River (DGPCE, 2019, Fig. 6b).





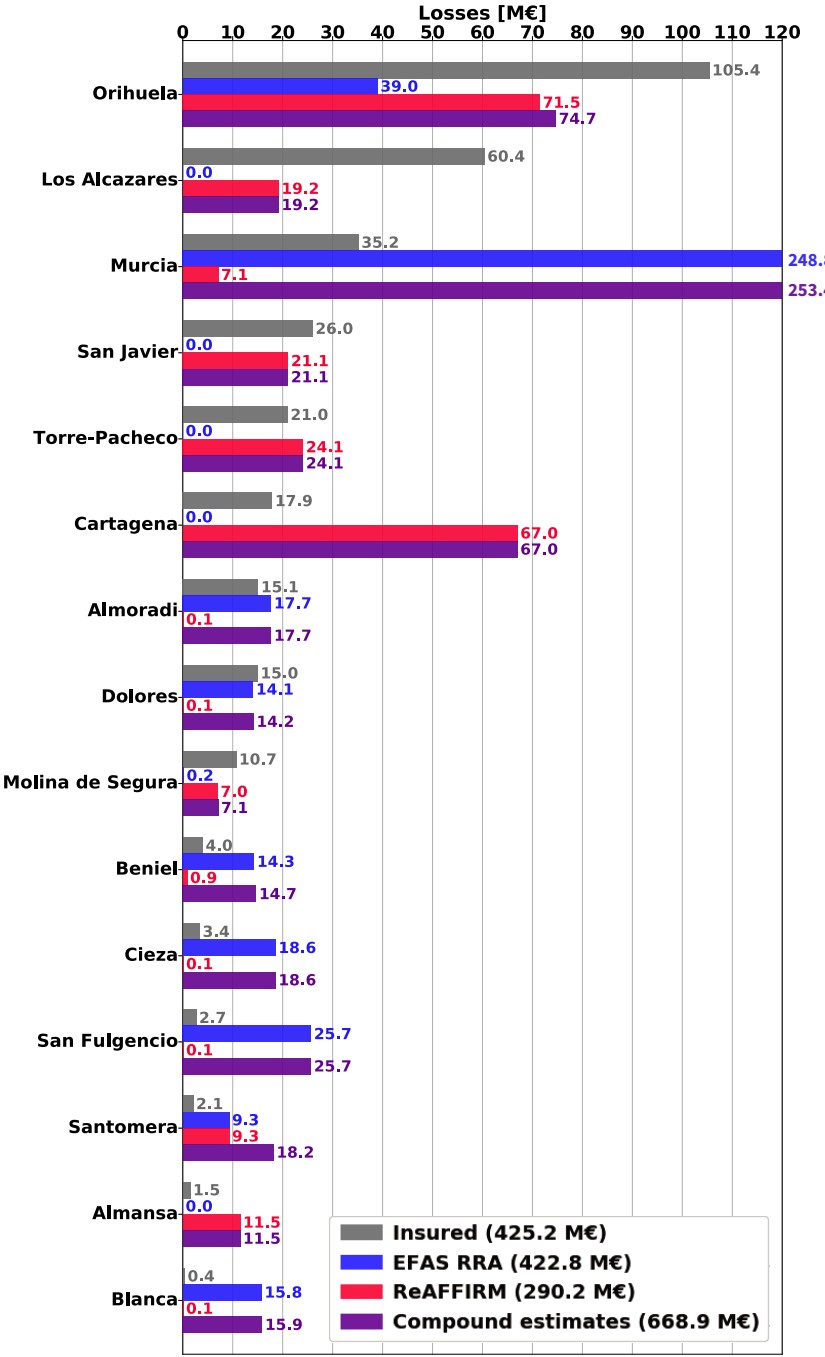

**Figure 9.** Comparison of insured losses (CCS, 2020) with those estimated by the two individual methods (Sect. 4.1 and 4.2) and the compound impact estimation (Sect. 4.3) in the 15 municipalities with insured or simulated losses greater than 10 million Euros (corresponding to Table 4).





To evaluate the simulated impacts also from a quantitative perspective, we have conducted a correlation analysis of the loss estimates with the insurance claim database (CCS, 2020) over the 907 municipalities in the domain (Table 5). The rank correlation coefficients of Spearman ($\rho$) and Kendall ($\tau$) have been used to avoid an overly strong penalisation by large differences in single data points (as e.g. in the Pearson correlation; see Croux and Dehon, 2010). These two coefficients measure to what degree the relationship between two datasets is monotonic, i.e. how well one variable can be expressed as a monotonic function of the other. Intuitively, values of $\rho$ or $\tau = 1$ correspond to a perfect correlation, whereas values of $\rho$ or $\tau = 0$ mean that the datasets are uncorrelated (Croux and Dehon, 2010). Values of $\rho > 0.5$ are commonly considered a strong correlation (see e.g. Couasnon et al., 2020; Titley et al., 2021) and values of $\rho$ are generally higher than those of $\tau$ (see e.g. Camus et al., 2021; Wahl et al., 2015). The results of $\rho$ and $\tau$ computed for our analysis show a moderate but significant correlation of the separate loss estimates from EFAS RRA and ReAFFIRM with the insurance claim database (Table 5). Furthermore, the correlation of the compound loss estimates with the insured losses is stronger than for those generated by the separate two methods (Table 5). This illustrates how the integration of EFAS RRA and ReAFFIRM into one compound flood impact estimation has improved the agreement between the simulated loss estimates and the reported insured losses.

**Table 5.** Correlation coefficients of economic losses (simulated by the individual methods and their combination) with the insurance claim database (CCS, 2020) in the 907 municipalities inside the domain.

| Method of loss estimation | Correlation with insurance claims | |
|---|---|---|
| | Spearman ($\rho$) | Kendall ($\tau$) |
| **EFAS RRA** | 0.41 | 0.38 |
| **ReAFFIRM** | 0.49 | 0.45 |
| **Compound estimation** | 0.55 | 0.51 |

## 5   Conclusions

This study proposes a more integrated perspective of flood early warning. Flood forecasting approaches are commonly designed individually for the different physical processes inducing flooding (i.e. separate systems for fluvial, pluvial, coastal, and flash floods). Especially during compound flood events, the monitoring of these separate systems can be time-consuming and challenging for emergency managers, potentially leading to a delayed and suboptimal emergency response. We propose to improve the current decision support by integrating existing flood type-specific impact forecasting methods into an overall compound flood impact forecast. This idea has been explored by combining real-time adapted impact assessments for fluvial floods (using EFAS RRA Dottori et al., 2017) and flash floods (using ReAFFIRM Ritter et al., 2020a) for a recent catastrophic episode of compound flooding in Southeast Spain.

The two separate impact assessments have been merged using simple predefined criteria. Despite the simplicity of the approach, the generated compound impact estimates corresponded significantly better to the observed impacts than those generated by the two individual methods. The number of false negatives in the most affected municipalities was reduced to





zero through the combination of the methods, and the correlation of the simulated economic losses with insurance claims

was higher for the compound impact estimation than for the individual two methods. Apart from the increased accuracy, the integrated impact estimation improves the usability for the end-users: With the separate methods as decision support, it might not be fully clear to the end-users why the flood type-specific assessments show fundamentally different results (although from the scientific perspective it makes perfect sense). The presented integration of the two methods into one unified output may be easier to monitor and interpret, enabling a more immediate and effective emergency response.

The overall compound flood impacts simulated over the two analysed river basins corresponded well to the impacts reported by various validation sources. When analysing the results at smaller scales (e.g. the municipality level), the underlying uncertainties are more apparent. The most important sources of uncertainty affecting the performances of the two methods and their combination appeared to be the accuracies of the employed hydrometeorological inputs and flood maps. Similar as in previous studies, the quantitative estimation of economic losses has been subject to high uncertainties in both of the methods (Dottori

et al., 2017; Ritter et al., 2020a); however, the previously reported systematic overestimation of losses by ReAFFIRM has not been confirmed, likely due to the higher availability of flood depth data in the present case study area.

For the analysis carried out in this paper, the impacts simulated by EFAS RRA and ReAFFIRM have been aggregated over the full event duration and subsequently merged. Combining the two methods in an operational setting would require the merging of real-time outputs with different temporal resolutions and lead times (Table 1). One way to facilitate this task could

be the application of blended rainfall products from radar and NWP (as e.g. applied in the TAMIR project Niemi et al., 2021), as a common input for the two methods. However, the uncertainty in forecasts for flash floods are typically higher than for fluvial floods when considering longer forecasting horizons (e.g. days). Such aspects would need to be taken into account in an operational combination of EFAS RRA and ReAFFIRM.

The combined impact estimation for fluvial and flash floods presented in this study can be applied at regional scale. To

extend the approach to the European scale, ReAFFIRM could be replaced by the newly developed pan-European approach for assessing flash flood impacts, named ReAFFINE (Real-time Assessment of Flash Flood Impacts at paN-European scale; Ritter et al., under review). This continental method has also been applied for the event analysed in this study, and the results show a high correspondence with the regional flash flood impact estimates generated by ReAFFIRM (Ritter et al., 2021). Due to the coarser resolution of ReAFFINE, the combined product with EFAS RRA over Europe should be issued at the regional level,

rather than the aggregation at municipality level done in this study.

An alternative procedure to the simple merging of the separate impact estimates proposed in this study could be to first simulate the compound flood hazards (e.g. in terms of compound water levels), and then translate the compound hazards into impacts. This would likely improve the quality of the impact estimation, especially for situations in which the spatial overlap of different flood types plays a crucial role (e.g. combined fluvial and coastal flooding during hurricanes). Several

existing methods assess the compound water levels for different combinations of flood types (e.g. Apel et al., 2016; Chen et al., 2010; Santiago-Collazo et al., 2019), however, these have not yet been adapted to real-time conditions due to the high computational cost of the underlying coupled hydraulic models (especially when focusing on large spatial domains Bates et al., 2021). These computational constraints make the creation of a full compound flood hazard forecast seem unfeasible for the



near future. Meanwhile, simple combinations of flood type-specific impact simulations (as proposed in this study) represent a
sound solution for forecasting compound flood impacts.

While this study investigated the combination of fluvial and flash floods, future efforts should aim at solutions for integrating
also systems designed for pluvial floods and storm surges (in coastal areas). For pluvial floods, a few impact forecasting systems
were demonstrated at the scales of cities or small regions, expressing impacts e.g. in terms of affected population (Aldridge
et al., 2016) or land uses (Hofmann and Schüttrumpf, 2019), economic losses (Rözer et al., 2021), or qualitative impact levels
(Speight et al., 2018). For coastal floods, forecasts of impact indicators (e.g. the building-waterline distance Harley et al.,
2016) and economic losses Bolle et al. (2018); Ferreira et al. (2018) were proposed at local or regional scales. As can be seen,
these approaches estimate the impacts in partly different metrics than the two methods in this study (which assess the affected
population, critical infrastructures, and economic losses). To estimate the impacts in terms of the same quantitative categories,
some of the mentioned approaches could also employ the exposure and vulnerability datasets used in this study (available at
European scale; Sect. 3.1 and 3.2). This would enable a more straightforward integration into the presented compound flood
impact estimation.

The results obtained in this study demonstrate the potential of integrating flood type-specific systems into a compound flood
impact estimation for improving the decision support during floods. A long-term vision is to integrate also systems designed for
other weather-induced hazards (e.g. snowfall or windstorms) into impact-based multi-hazard EWSs. This development would
be a significant contribution towards a society that is more resilient to natural disasters (Merz et al., 2020; Rebora et al., 2019;
UNISDR, 2015b; WMO, 2015, 2018).

*Data availability.* Weblinks to the open datasets used in this study are included in the text body or the reference list. The authors do not hold
the rights to distribute the remaining data used.

*Author contributions.* This work was initialised through ideas and discussions involving all of the authors. Following these discussions, JR,
FS, and MK conceptualised the study. MK carried out the simulations of EFAS RRA, while JR applied the ReAFFIRM method in the study
area for the analysed flood event. JR, MB, FS, and MK analysed the simulation results. Finally, JR drafted the original manuscript, which
was then reviewed by the co-authors. DS was responsible for funding acquisition and the provision of resources, and MB and DS supervised
JR throughout the development of this study and his doctoral dissertation.

*Competing interests.* The authors declare that they have no conflict of interest.

*Acknowledgements.* We would like to thank OPERA, the Spanish State Meteorological Agency (AEMET), the Valencian Association of
Meteorology (AVAMET), and the Hydrographic Confederation of the Segura River (CHS) for the provision of hydrometeorological data.





The Spanish National Geographic Institute (IGN) kindly granted access to the topography, flood maps, and land use datasets used for applying ReAFFIRM in the study area. We extend our gratitude to the Spanish Insurance Compensation Consortium (CCS), the Spanish Directorate-General for Civil Protection and Emergencies (DGPCE), the Copernicus Rapid Mapping Service, and various news media for reporting the impacts of the DANA event. Lastly, special thanks are owed to Shinju Park and Calum Baugh for fruitful discussions and the provision of information and data.

The Horizon 2020 project ANYWHERE (H2020-DRS-1-2015-700099) financed the initial period of this study and a four-month visiting stay of JR at the European Commission Joint Research Centre in Ispra (Italy). The study has been finalised in the frame of the TAMIR project (UCPM-874435-TAMIR).





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
