# Peer review of "Compound flood impact forecasting: Integrating fluvial and flash flood impact assessments into a unified system"

_Hydrology and Earth System Sciences, 2021_

## Author Comment (AC1)

**1. Response to Reviewer 1 (Mario Rohrer)**

We would like to thank Mario Rohrer for the time and effort spent on reviewing this manuscript, and for his positive and constructive feedback. Please find below the details of the modifications we have introduced in response to the comments.

> **1.1.** *Abstract, Lines 7ff: "…, this paper proposes the integration of different flood type-specific approaches into one compound flood impact forecast. This possibility has been explored by combining the simulations of two impact forecasting methods (representing fluvial and flash floods) for a recent catastrophic episode of compound flooding:" I think it's important to mention already in the abstract on what existing products your proposed impact forecast is based. Suggestion:*
> *…, this paper proposes the integration of two flood type-specific approaches (representing fluvial and flash floods) into one compound flood impact forecast. For this scope a 'unified system' was developed by combining the simulations of two impact forecasting methods: One based on the European Flood Awareness System (EFAS), the other on flash flood hazard nowcasts obtained with the European Rainfall-Induced Hazard Assessment (ERICHA) system. This possibility has been explored by combining the simulations for a recent catastrophic episode of compound flooding:…*

Thank you for this nice suggestion. We agree that it is important to mention already in the abstract the employed methods. We have adopted the suggested phrasing with a few minor adjustments. The first mentioned sentence, we prefer to keep general, as this paper does not only propose the combination of forecasts for fluvial and flash floods, but the combination of flood impact forecasts in general (also including other flood types). Fluvial and flash floods are used in this paper as an example to illustrate the benefits of integrating forecasts of different flood types. In the revised manuscript, the paragraph will then read as follows:

> *"…, this paper proposes the integration of different flood type-specific approaches into one compound flood impact forecast. This possibility has been explored through the development of a unified system combining the simulations of two impact forecasting methods: the Rapid Risk Assessment of the European Flood Awareness System (EFAS RRA; representing fluvial floods) and the radar-based ReAFFIRM method (representing flash floods). The unified system has been tested for a recent catastrophic episode of compound flooding: …"*

> **1.2.** *Lines 10-11: "the DANA event of September 2019 in Southeast Spain." For the non-Hispanic reader, it may not be clear what DANA means. Suggestion:*
> *the DANA (Depresión Aislada en Niveles Altos, Cut-off Low) event of September 2019 in Southeast Spain.*

We have included the explanation of the acronym in the revised abstract.

> **1.3.** *Line 14: "Although the compound impact estimates were less accurate at municipal level, they corresponded significantly better to the observed*

*impacts ...". It's only one case, thus 'significantly' may not be adequate, I would say: MUCH better. Suggestion:*
*Although the compound impact estimates were less accurate at municipal level, they corresponded much better to the observed impacts...*

Agreed and changed.

**1.4.** *Introduction, Line 85: "has been taken as an opportunity to explore the possibility of such an integrated system". I think to make a system more complex may always also imply some disadvantages/drawbacks. Suggestion:*
*has been taken as an opportunity to explore the possible advantages and drawbacks of such an integrated system.*

Thanks for this suggestion, we have adopted this phrasing.

**1.5.** *Perhaps you can explain a little bit more in detail what a DANA-event is: see e.g. Ferreira, 2021; Garcia-Ayllon, S.; Radke, J., 2021; Giménez-García et al., 2021.*

We have included some more information on the DANA phenomenon and a reference to the interesting paper of Ferreira et al. (2021). Thanks for pointing out this recent work. The paragraph introducing the DANA phenomenon reads now as follows:
*"From 11 to 15 September 2019, a weather phenomenon commonly known in Spain as "DANA" or "Gota Fría" (MartínLeón, 2003) affected the south-eastern part of the country. The term DANA means "upper tropospheric cut-off low", a situation occurring typically in autumn when easterly winds push warm humid air masses from the Mediterranean Sea towards the steep topography of the coastal region (Ferreira, 2021)."*

**1.6.** *Fig. 2. Concerning the legend: I suppose the colors are representing the RETURN PERIOD! If this is the case, please write it! This is a very nice figure, but I don't see how the reader can compare the return period at a gauging station with a peak flow runoff in m3/s. Perhaps you can calculate a return period of the runoff gauges, if not, perhaps you can indicate the rank of the runoff, or a similar metric which is in a way comparable to a return period.*

We have added to the legend of Figure 2 the missing term "return period".
The selection of the EFAS flood maps along the Segura is based on comparing the measured peak flows to the input discharges used for the hydraulic simulations of the flood maps (see lines 152-155). To make this connection clearer, we have added to the labels of the stream gauges in Figure 2 the discharges that were used as input for the hydraulic simulation of the selected flood map. The caption of the Figure has been adjusted accordingly:

[Figure]

Figure 2. Peak flows measured at the gauging stations in the Segura River during the DANA event, and (in brackets) at each station the input discharge of the most closely corresponding EFAS flood map. The 5 km-grid cells represent the resulting selection of EFAS flood maps along the LISFLOOD drainage network. Map data ©Google Earth 2015.

*1.7.* *Perhaps you should mention in the conclusion that this is a case study and that is important to try this method also for other extreme large events as for example the event of 01.09.2021 over Castilla to explore better the advantages and drawbacks of the proposed product.*

Yes, this is an important point. We have modified the related statement in the conclusions (lines 426ff) and included the need for further testing of the approaches on other compound flood events.

---

## Author Comment (AC2)

**2. Reviewer 2**

We would like to thank the reviewer for the time and effort spent on reviewing this manuscript, and for his positive and constructive feedback. Please find below the details of the modifications we have introduced in response to the comments.

> ### 2.1. *How to account for uncertainties deriving also from the integration of flood-type specific forecasts? Even if it is not the purpose of this paper could you provide some ideas on how to account for the uncertainties resulting from the combination of the two separate approaches (EFAS RRA and ReAFFIRM)?*

As described in section 3.3 and in the conclusions in lines 386-388, the chosen approach to combine the impact estimates from the two methods is very simplistic. As pointed out by the reviewer, this simple way of combining the estimates introduces additional uncertainties into the resulting compound impact estimates. For instance, some areas with simulated fluvial flood impacts might in reality be affected also by flash floods with potentially different damaging mechanisms. As described in the conclusions in lines 416-419, an alternative to mitigate these uncertainties could be to use coupled hydraulic models for combining the methods already at the stage of the hazard estimation (e.g. in terms of compound water levels), before translating these jointly into socio-economic impacts. As of today, however, running such coupled hydraulic models in real time is not feasible, mostly due to computational constraints (as described in lines 419-425).

> ### 2.2. *Which are the most relevant sources of uncertainty that can be associated to the ReAFFIRM methodology? Maybe it is reported in previous papers from the same author but a short discussion should be also reported here.*

Yes, that is indeed important to mention. We have added the following sentence to the description of ReAFFIRM in section 3.2 (before line 201):
*"As discussed in detail by Ritter et al. (2020a), the most pronounced sources of uncertainty affecting the ReAFFIRM impact estimates are the qualities of the employed rainfall inputs and flood maps. Additional important uncertainty sources include the purely rainfall-based hazard estimation and the vulnerability datasets used for translating flood hazard into socio-economic impacts."*

> ### 2.3. *How the uncertainties in the flood estimation can be translated into uncertainties in impact estimation? I think this process is different from flood and flash-flood processes. A short discussion related to this issue should be added.*

For both methods, the uncertainties in the flood hazard estimation propagate down to the impact estimates. In ReAFFIRM, some of the uncertainties in the flood hazard estimation are reflected by the lower and upper bounds of estimated impacts (see lines 201-203 and for greater detail Ritter et al., 2020a). To provide some more information on this, we have added a summary of the most significant uncertainty sources along the ReAFFIRM model chain (see the previous comment).

In EFAS RRA, the flood hazard and thus the impacts are estimated deterministically (using as input the ECMWF ensemble median, see Table 1). A fully probabilistic estimation of impacts would require running the methods in real time on numerous ensemble members, which is – given the computational requirements and the need for fast generation of the warnings – currently not feasible. In the meantime, the simple deterministic impact estimation presented in this paper offers a sound solution.

**REFERENCES**

Ritter, J., Berenguer, M., Corral, C., Park, S., and Sempere-Torres, D.: ReAFFIRM: Real-time Assessment of Flash Flood Impacts – a Regional high-resolution Method, Environment International, 136, 105 375, https://doi.org/10.1016/j.envint.2019.105375, 2020a.

---

## Author Comment (AC3)

**3. Reviewer 3**

We would like to thank the reviewer for the time and effort spent on reviewing this manuscript, and for his positive and constructive feedback. Please find below the details of the modifications we have introduced in response to the comments.

> *3.1.    The methods employed in the manuscript are intended for civil protection and emergency services. It would be interesting that the authors recall the needs of these authorities in that field for the two flood types addressed in the manuscript : fluvial floods and flash-floods.*

The needs of the end-users are similar across flood types, and we have added a sentence summarising the most important requirements in line 25ff (which described in detail by the added WMO reference):

> *"To enable an effective emergency response, the warning information needs to be accurate, easily interpretable, and disseminated in a timely manner to end-users such as civil protection authorities (WMO, 2018b)."*

A special requirement during flash floods is that the delay of the warnings must be extremely short, as the fast-evolving nature of such events leaves only little time to react. We have added the following sentence after line 35:

> *"The fast-evolving nature of flash floods require a quick computation and dissemination of the warnings to the end-users to maximise the time available for emergency response measures (e.g. evacuations or road closures)."*

> *3.2.    The reading of the manuscript would be easier if the authors presented how these methods can be used in an operational and "real-world" context according to the type of flood.*

To point out more clearly the use of the flood forecasts in the operational work of the end-users, we have added in the introduction before line 41 the following sentence:

> *"The hazard forecasts provide information of potential flood locations and magnitudes before the onset of the event and thus help to coordinate measures such as warnings or evacuations."*

Furthermore, regarding the past usage of EFAS RRA in operational settings, we have expanded the sentence in line 56ff to the following:

> *"As part of the European Flood Awareness System (EFAS), the RRA has been providing for a few years operational decision support to various end-users across the continent, who monitor the outputs on a daily basis for the coordination of response measures in case of emergencies."*

> *3.3.    It is not clear if the assessment of flood impacts is performed from forecasts, simulations or observations.*

To minimize external uncertainties for the purpose of this study, both methods use as inputs hydrometeorological observations (no forecasts), as pointed out in lines 123-125 and in Table 1.

*3.4.      It would have interesting to study the sensitivity of the obtained to forecast uncertainties, especially for flash-floods where these uncertainties are often very large.*

Yes, it would indeed be very important to analyse how the larger uncertainties induced by the forecast propagate down to the impact estimates, but such a sensitivity analysis is unfortunately out of the scope of this study. To point towards this direction, we have added a remark on this in the conclusions. Lines 406ff will then read as follows:

*"However, the uncertainty in forecasts for flash floods is typically higher than for fluvial floods when considering longer forecasting horizons (e.g. days), and the sensitivity of the impact outputs towards the increased uncertainty in the forecast inputs requires further investigation."*

*3.5.      As noticed by the authors, a fluvial flood and a flash flood display very different dynamics which could result in different applications conditions. How the authors deal with this point to estimate of a compound flood?*

Although in this study, the impact estimates are presented summarised over the full event duration, in a real-time application, the compound impact estimates/forecasts would be computed time step by time step. For instance, ReAFFIRM computes the flash flood impacts every 15 minutes, while EFAS RRA the river flood impacts every 6 hours (see Table 1). The real-time compound impact output would be merged every 15 minutes, using the current ReAFFIRM outputs and the most recent (6-hourly) EFAS RRA outputs. In this way, the compound output would display the different dynamics of both flood types. The conclusions (lines 402-408) provide an outlook on the necessary steps for a real-time implementation of the compound flood impact estimation, taking into account differences between the flood types, e.g. with respect to the lead times.

**REFERENCES**

WMO: Multi-hazard Early Warning Systems: A Checklist. Outcome of the first Multi-hazard Early Warning Conference, Tech. rep., WorldMeteorological Organization, Cancún, Mexico, 2018b.